# Im2Contact: Vision-Based Contact Localization Without Touch or Force Sensing

**Leon Kim, Yunshuang Li, Michael Posa, and Dinesh Jayaraman**
GRASP Laboratory
University of Pennsylvania, USA
{leonmkim, sheylali, posa, dineshj}@seas.upenn.edu

**Abstract:** Contacts play a critical role in most manipulation tasks. Robots today mainly use proximal touch/force sensors to sense contacts, but the information they provide must be calibrated and is inherently local, with practical applications relying either on extensive surface coverage or restrictive assumptions to resolve ambiguities. We propose a *vision*-based extrinsic contact localization task: with only a single RGB-D camera view of a robot workspace, identify when and where an object held by the robot contacts the rest of the environment. We show that careful task-attuned design is critical for a neural network trained in simulation to discover solutions that transfer well to a real robot. Our final approach `im2contact` demonstrates the promise of versatile general-purpose contact perception from vision alone, performing well for localizing various contact types (point, line, or planar; sticking, sliding, or rolling; single or multiple), and even under occlusions in its camera view. Video results can be found at: https://sites.google.com/view/im2contact/home.

**Keywords:** contact perception, manipulation, vision-based

## 1 Introduction

Perceiving and reacting to contact is critical for performing manipulation tasks [1–5]. Consider what happens when a person puts a book on a crowded shelf: they hold the book and aim for a gap until it meets resistance, jostle the book to make room, then press sideways to line up the book against its neighbor, and finally slide the book snugly into place. Throughout, the key events they must track all have to do with the physical contacts between various surfaces: when and where they are made, broken, and transition from sticking to sliding. What means does a robot have today to sense such contacts between its body, its tools, and external objects?

Current contact perception techniques for robots operate mainly from force torque and touch sensing. Force torque (F/T) sensors [6] located at the robot's joints can inform model-based contact estimation techniques [7–11]. However, this estimation problem is under-determined. To see this, consider a two-fingered manipulator in two contact configurations: applying a 10 N force to one finger, versus applying a 5 N force to each finger. A wrist F/T sensor located directly behind the gripper's midpoint is aliased: it senses identical forces in both cases. For such direct contacts with the robot, aliasing can be partially resolved with proximal touch sensors [12–15] applied to contacting surfaces on the robot, but over only a limited area. Further, consider as in our introductory example, a robot holding a book that contacts a bookshelf. Such "extrinsic" contact is still only sensed indirectly at the fingers, and remains impossible to resolve. To overcome these issues, today's contact estimation techniques reduce the number of unknowns by operating under restrictive assumptions, such as about the number and types of contacts, and the shapes of the various objects involved (Sec 2). Finally, even when operating within these assumptions, F/T and touch sensors are often expensive, and drift or deteriorate quickly over use [15–18] requiring frequent and cumbersome re-calibration.

7th Conference on Robot Learning (CoRL 2023), Atlanta, USA.

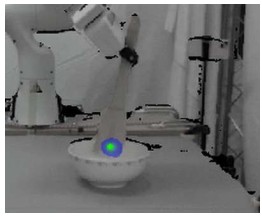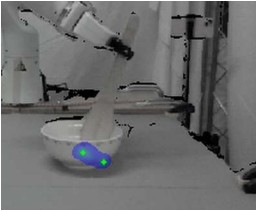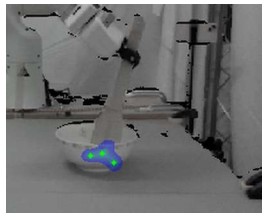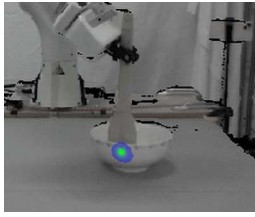

• Prediction  ● Probability > ε

Figure 1: Sequence of predictions of our method `im2contact` for localizing extrinsic contact in an image between a grasped object (such as a spatula) and the environment (such as a bowl)

To address both the lack of global information available to F/T and touch sensors, and the sensor drift issue, we instead operate from camera images. Cameras can see the entire scene, and function without deterioration for long durations. Since contacts are a function of the shapes and movements of the objects and the environment, contact estimation would be easy if we could first get perfect 3D shape and pose estimates from vision. However, we argue that contact from 3D would be a poor solution. First, given occlusions around the critical contact regions and arbitrary-shaped objects, near-perfect shape and pose estimates would require impractically many camera views. Next, the often-crucial binary distinction between near-contacts and true contacts can be hard to capture by continuous-valued object pose estimates.

Rather than infer contacts from intermediate visual representations such as shapes and poses, we propose `im2contact`, a more direct approach for data-driven, model-free visual contact location estimation at the outputs of a deep neural network. Our system is trained entirely in simulation and incorporates the combination of cropped depth images of the salient regions of the scene, an additional reference depth image to specify the grasped object, and motion cues from optical flow.

In our zero-shot transfer evaluations on a real robot, `im2contact` models demonstrate the possibility and promise of versatile general-purpose contact perception from vision alone, performing well for localizing various contact types (point, line, or planar; sticking, or sliding; single or multiple), and even under occlusions in its camera view.

## 2   Related Work

The robotics community has long recognized the crucial role of external contact sensing in manipulation, particularly of contacts between the manipulator and environment (e.g. [8, 9, 19]), coined by Bicchi et al. [7] as "intrinsic contact". However, "extrinsic contact," e.g. between a held object and the environment, is similarly useful, but notably more difficult. Prior work on extrinsic contact has primairily focused on the use of force and touch sensors and typically found success through strong assumptions which ultimately limit the potential scope of the results. For example, related work has relied on the assumption and/or enforcement of pre-defined contact configurations [20–22], limiting application to grasped objects seen during training [23], or tight coupling with information gathering motions [24]. Other approaches have incorporated pointclouds with force and touch sensing, but assume full coverage of grasped objects [25, 26] or restrict interactions to line contacts [27].

In order to enable robots to readily use unmodeled tools in unstructured environments, many of the above restrictions must be lifted. We make steps towards this via the choice of vision-based sensing with no explicit assumptions made on possible contact configurations, object properties, and minimal access to privileged knowledge of the object or environment.

## 3   Visually Localizing Extrinsic Contacts In General Manipulation Scenes

For a robot arm holding a grasped object, we are interested in localizing contact between the held object and the rest of the environment, as the robot arm moves in its workspace. Unlike much of the

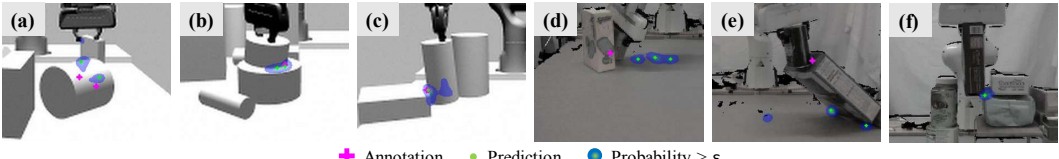

➕ Annotation    • Prediction    🔵 Probability > ε

Figure 2: U-Net Depth generally performs well in sim but poorly on real data in challenging cases such as: **(a,d)** occluded contact **(b,e)** ambiguous grasped object geometry **(c,f)** near-contact with background object

prior work described above, we do not assume access to prior information about the held object or the environment, other than the table.

As inputs at each time instant, we assume a H×W view of the robot workspace from a single fixed RGB-D camera as shown in Fig 3, alongside proprioceptive sensing of the robot state from joint encoders. This is a minimal sensing setup for vision-based robot manipulation, chosen to maximize the scope of our problem formulation.

As output, we would like our system to generate a H×W map of estimated contact locations, that can be overlaid on the camera image, similar to dense image-based features [28, 29]. To achieve this, we will train neural networks with standard pixel-wise binary cross-entropy classification objectives. This treats the the output contact map values at each pixel as contact probabilities, and maximizes the likelihood under the model of the annotated ground truth contact locations in the training data.

### 3.1 Simulated Training Data

It is not practical to obtain ground truth contact location annotations from real video, but fortunately simulators provide this information. To generate target contact maps for training, we project 3D contact points from the Gazebo simulator [30] into the camera frame. Next, we generate training data in an episodic fashion. In each 15-second episode, we randomize the geometry, masses, friction coefficients, and initial poses of grasped and environment objects. Shapes are chosen to be cylinders, spheres, and cuboids with random parameters. The grasped object is rigidly attached to the robot arm throughout the episode, as the robot end-effector moves to randomly set targets with a low-impedance controller to generate rich interaction data. Additional details are included in Appendix A. Our 4500 episodes (675000 frames at 10 fps) of training data span many types of extrinsic contacts: point, line, and plane contacts, instantaneous collisions and sustained sliding or rolling contacts along with simultaneous contact with multiple bodies.

### 3.2 A Simple Baseline To Illustrate The Difficulty of Sim-To-Real Transfer

To motivate our final method, we first showcase the difficulty of sim-to-real transfer with a baseline, U-Net Depth. U-Net Depth builds on the widely used U-Net [31, 32] architecture for image-to-image problems. In the U-Net, an encoder first assimilates information from over the entire 240×320 input image into a "bottleneck" representation of size 15×20×1024. Then, a decoder iteratively spatially upsamples this representation with the aid of skip connections from intermediate encoder layers, to finally produce an output over the original 240×320 input dimensions. To facilitate sim-to-real transfer and access 3D information, U-Net Depth uses depth images [33] as input rather than color images. We train using pixel-wise cross entropy.

U-Net Depth trains to near-zero training losses, and performs very well on held-out data in simulation. However, on real data, its performance deteriorates significantly. In Fig 2, we anecdotally note a few challenging scenarios where this baseline struggles in the real world: (a) under significant occlusion, it misses the occluded contact and predicts irrelevant false positives on the table, (b) when the geometry of the grasped object is ambiguous, it predicts contact between an extraneous box and table, and (c) it produces far more false positives in near-contact scenarios than in sim. We more thoroughly evaluate U-Net Depth in Sec 4.

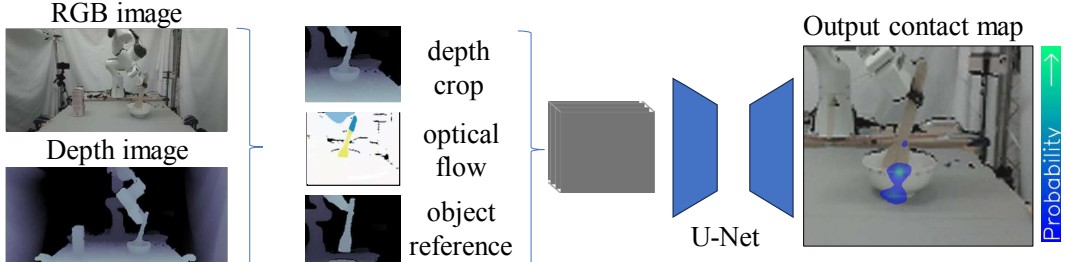

Figure 3: `im2contact` architecture. Depth and flow are both cropped and concatenated before being passed through the U-Net. The object reference is passed through a separate encoder whose output is concatenated at the bottleneck of the U-Net. The final output of the U-Net is a probability map over the original image dimensions.

### 3.3 Facilitating Generalizable Contact Localization With `im2contact`

To mitigate the failures of U-Net Depth on real data, we consider the possible causes of poor generalization. Machine learning systems commonly over rely on spurious correlates of the target labels in the training data [34–36], which fails to generalize under distribution shift. During training, spurious correlates could distract the optimizer [36] from finding true "causes"; this can be alleviated by anticipating such distracting but irrelevant correlates in advance and removing them from the learner's inputs. Inputs might be also incomplete, lacking information on the true underlying cause of the outputs. We propose three potential improvements to U-Net Depth, one to remove distracting correlates, and two to add missing causal information into the inputs.

- **Depth Cropping (+ crop):** Assuming grasped objects that are not very large, all extrinsic contacts will occur close to the end-effector. With a calibrated camera and proprioceptive knowledge, it is easy to locate the end-effector in the image. We propose to focus the network on the most relevant regions by cropping the depth images into a 90x110 box around this end-effector location at each time, before feeding them to the U-Net. In addition, we add three channels for coordinate convolution as proposed by Liu et al. [37] to provide the network with the spatial relationship of the cropped image with respect to the original image.

- **Grasped Object Reference (+ obj-ref):** We have thus far reduced contact localization to an instantaneous task, performed from the current sensory observations. However, consider the scenario when the grasped object is occluded by the environment: if we do not know its spatial extent, it may be impossible to know whether there is any extrinsic contact. Similarly, in cluttered views it may be difficult to disambiguate between the boundaries of the grasped object and the environment from a single depth image. We propose to provide such missing information through an additional input: a single reference depth image of the grasped object in the robot gripper, partially specifying the shape of the grasped object.

- **Optical Flow (+ flow):** From a single image, it may be difficult, or even impossible, to differentiate between contact and near-contact. For example, a gap between the grasped object and the environment of a few millimeters is likely imperceptible without a perfect vantage. However, contact forces often induce motion in the environment, causing environmental objects to slide or roll. To capture these cues, we propose to use optical flow computed from the camera RGB images.

In the rest of the paper, we use the name `im2contact` for the combined approach: U-Net Depth + crop + obj-ref + flow. Fig 3 schematically depicts the network with all inputs and outputs.

**Reducing To Discrete Contact Locations.**   Our training procedure generates contact probability maps, but it is convenient for evaluation and useful for downstream tasks to identify discrete contact locations. To this end, we adapt the greedy non-maximum suppression (NMS) technique [38–40], commonly used in object detection. To identify spatially separated peaks in the contact probability

map, we first reject all pixels scoring below $\epsilon = 0.01$, sort pixels by descending score, and iteratively reject pixels within a 5 px neighborhood of higher-scoring pixels. See Appendix B for pseudocode.

**Implementation Details.** The pixel-wise cross-entropy loss treats each pixel independently of all others and hence penalizes a 1 px deviation in predicted location equivalently to a 100 px deviation. To abate this, we spatially blur ground-truth contact maps before computing our training target, and find that this accelerates training. We monitor performance on held-out simulation data to implement early termination. For computing optical flow, we use the off-the-shelf RAFT [41] model. Code and models will be available at our website.

# 4 Experiments

We test `im2contact` in simulation and in real to evaluate performance in realistic tabletop manipulation settings. We further test, via ablation studies, whether, and under what settings, our proposed changes (crop, obj-ref, and flow) do improve real performance. We include additional anecdotal experiments which push `im2contact` beyond its training settings, evaluating generalization.

**Real Robot Data, Annotations, and Performance Metrics.** We perform teleoperated experiments in a table-top environment with a Franka Emika Panda robot arm and an Intel Realsense L515 RGB-D camera, with reasonably well-matched simulation and real robot setups, as shown earlier in Sec 3. To facilitate some quantitative comparison, we manually annotate contact locations for 30 episodes (approximately 13 mins comprising 12,362 frames, of which 1/3 involve at least one contact) of real robot interaction data, spanning variations in the grasped object, environment objects, robot movements, and contact scenarios. Recall that for simulation experiments, annotated contacts are readily available as in Sec 3; we use 500 episodes (75,000 frames) for computing simulation performance metrics.

Our main metrics are precision, recall, and their harmonic mean, the F1 score. All three must be in $[0, 1]$, and higher is better. To compute them, we first follow the procedure from Sec 3.3 for reducing the network's output probability maps to discrete contact locations. We then match each ground truth contact point to a predicted contact point with the Hungarian algorithm, and drop predictions where the ground truth is over 15px away as false positives. For our metrics, ground truth contacts with matches are true positives, and those without are false negatives. We run 5 random seeds for each method and report means and standard errors.

**Quantitative Results on Simulation and Real Data.** Fig 4 plots contact localization precision and recall in real (hollow circles) and simulated (solid circles) test data, for `im2contact`, the baseline U-Net Depth (Sec 3.2), and leave-one-out ablations of `im2contact` that in turn drop crop, obj-ref, and flow. Tabular results are presented in Appendix C.

First, for the simple U-Net Depth baseline, these results clearly validate our qualitative observations from Sec 3: precision and recall both deteriorate dramatically from sim to real. Next, U-Net Depth and `im2contact` are both among the best performing methods on sim data, but `im2contact` only drops marginally in performance on real data, so it remains among the best-performing methods in real data, while U-Net Depth performs worst.

Moving on to the leave-one-out ablations, "w/o crop" deteriorates nearly as much as U-Net Depth, while "w/o obj-ref" and "w/o flow" degrade more gracefully upon transfer to real data, crop contributes the most among the three components of `im2contact`. Fig 4 (middle) plots standard error for real data and the legend (right) lists F1 scores. From this initial coarse analysis with limited quantitative metrics, losing obj-ref marginally hurts precision, and losing flow does not significantly affect these scores. Note that these aggregate scores might not reflect performance in rare scenarios, of which there are many.

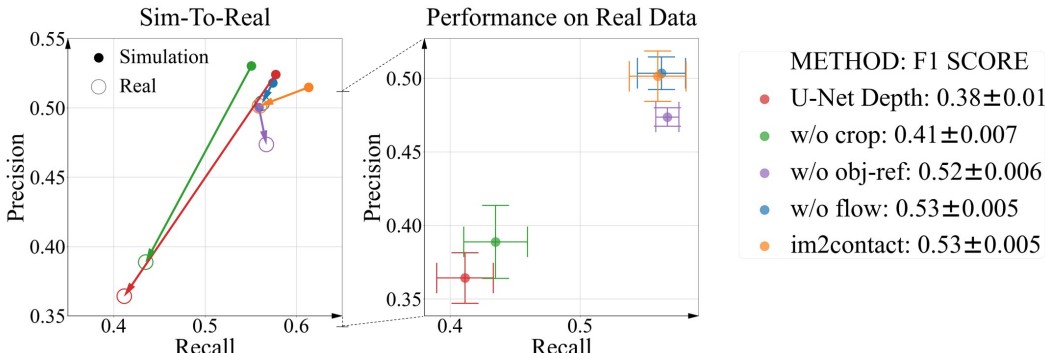

Figure 4: **(Left)** Degradation under sim-to-real transfer **(Middle)** Precision-Recall on real data **(Right)** Legend along with F1 scores on real data.

**Qualitative Analysis.** Armed with the initial evidence from the coarse quantitative results, we performed a thorough qualitative analysis over the aforementioned 30 real robot episodes. Figs 5 and 6 present keyframes of selected video reels. Fig 7 compares the outputs of all methods at some selected frames across the dataset. These examples are selected to illustrate some key insights from our more comprehensive analysis (remaining examples available on our website):

- As our metrics above suggested, crop is indeed the most important contributor to im2contact performance. In Fig 7, both im2contact "w/o crop" and U-Net Depth perform similarly poorly, demonstrating the criticality of focusing the model on relevant regions for finding good solutions.

- The grasped object reference frame (obj-ref) frequently improves performance in occluded contact situations (Fig 5, left), or when the grasped object moves coherently with others, creating a misleading flow field (Fig 5, right and Fig 7 (d)). This is consistent with our arguments in Sec 3.3; in such situations, knowing the shape of the grasped object is critical for localizing extrinsic contacts.

- Consistent with the metrics above, the effects of optical flow are less obvious, and w/o flow is very similar to im2contact in Fig 7. We have however anecdotally observed that it improves

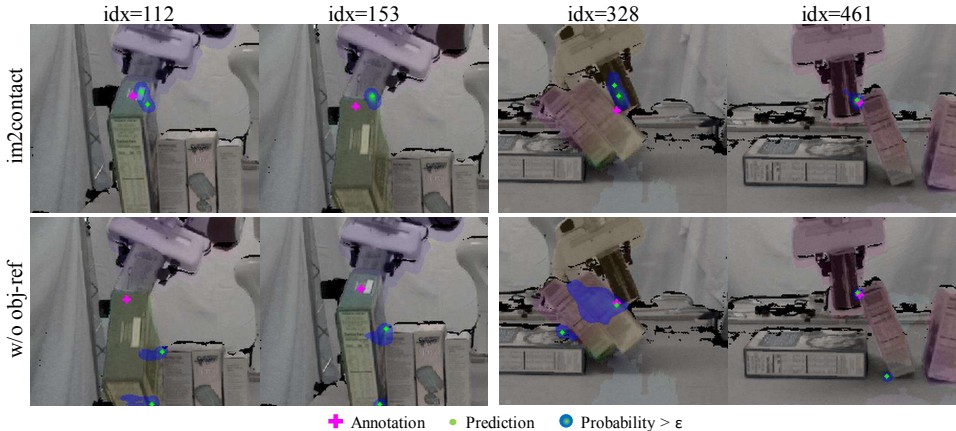

Figure 5: im2contact **w/o obj-ref**: obj-ref provides useful information about the grasped object's shape, which is not otherwise available under occlusions or misleading flow fields. Optical flow fields overlaid on all images. **(Left)** The grasped can behind the cereal box, slides up its side to topple the box. Here, the grasped object is occluded. **(Right)** The robot pushes two boxes to topple them, then pushes one of them back upright. These sustained pushes lead to coherent flow fields that do not help to separate a grasped object from a neighbor. In both cases, "w/o obj-ref" errs by treating the can-cum-boxes collective as a single grasped object, generating false positives at locations where the boxes contact their neighbors.

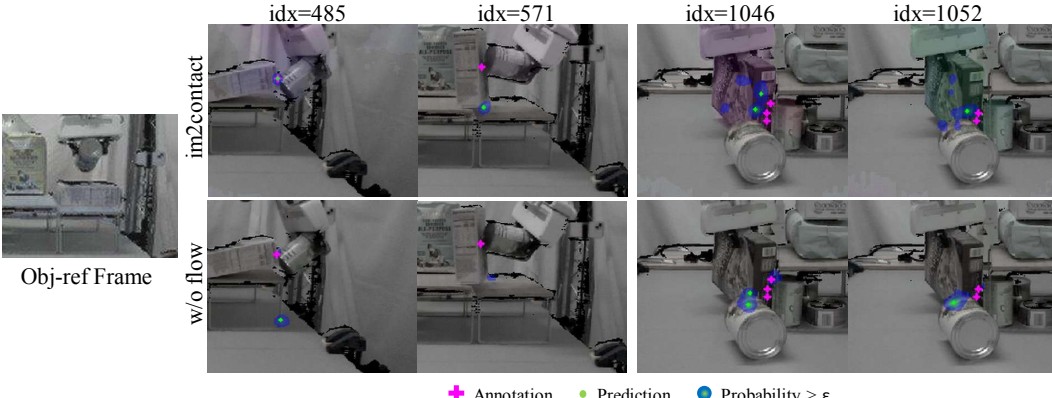

Figure 6: `im2contact` **w/o flow** (Optical flow fields overlaid only on `im2contact` images): **(Left)** The robot uses a grasped can to press against and pivot a box into an upright position. The reference frame does not clearly show the can's shape, so flow must separate the grasped object from others. Sure enough, in frame 485 during the motion, "w/o flow" predicts irrelevant contacts at the foot of the shelf, but `im2contact` performs correctly. Later, in frame 571 immediately after the motion, `im2contact` no longer has the flow cue, and fails. **(Right)** The robot uses a large box to sweep a number of cans under a shelf. In this highly cluttered scene, "w/o flow" consistently predicts a false contact between a stationary can in the foreground rather than the true contact happening in the background underneath the shelf. `im2contact` on the other hand uses the flow on the moving background can to localize the true contacts.

performance in certain cases. In Fig 6 (a), flow suppresses a false positive in a near-contact situation and helps identify a contact with a can within clutter in Fig 6 (b).

**Out-of-Distribution Evaluation of** `im2contact` We conclude with a brief highlight reel in Fig 8 visualizing the output of `im2contact` on chosen out-of-distribution settings that are difficult to annotate manually. Despite significant distribution shift from the training domain, we find `im2contact` still produces reasonable estimates in examples including moderately deformable objects and human demonstrations. The promising results on the latter suggest our extrinsic contact estimates may be amenable for use in guiding robot policy learning from human videos, previously demonstrated by Bahl et al. [42] in the context of intrinsic contacts. The details of our human demonstration evaluations are included in Appendix E and additional examples are on our website.

## 5   Limitations and Future Work

Our method, `im2contact`, using only a single camera and proprioception, can localize contacts in the 2D image. While this is already surprising and useful, there is much information that our system does not capture: it does not localize contacts in 3D, perceive contact forces, or classify modes of contact. Our deliberately simple sensing setup might be fundamentally limited for solving these broader problems. For example, predictions over the RGB-D image may be backprojected into 3D spatial coordinates using the depth map. However, single-viewpoint depth images are only a 2.5D representation of geometry, so we are unable to localize full 3D coordinates of contact points during occluded contacts. Multiple cameras may help address this issue.

In addition, our cropping method assumes an upper bound on the dimensions of the grasped object. Integrating an off-the-shelf object segmentation method may enable more more adaptive schemes for cropping and grasped object specification that may also improve robustness to occlusions.

Lastly, our method leverages the global information provided by vision over more local tactile sensing. In doing so, we sacrifice the precision of our method which is prone to false positives during near-contact and occlusions, even with the addition of optical flow. Our preliminary efforts to inte-

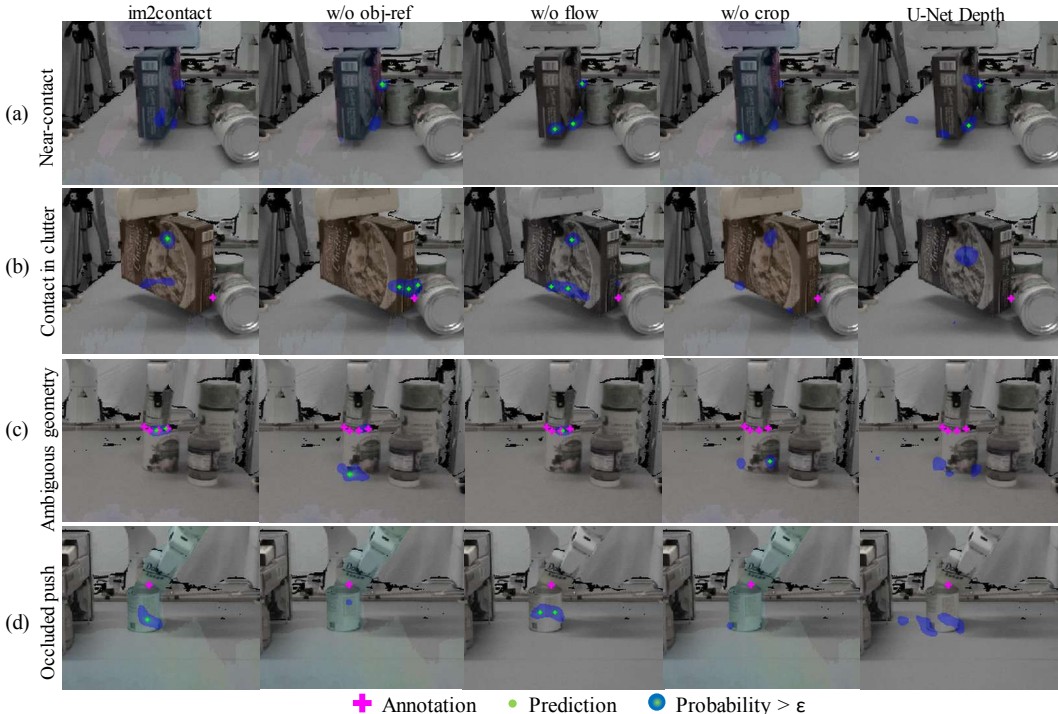

Figure 7: Our full ablations on four illustrative samples (flow visualized on the models that use them): **(a)** a grasped cereal box encounters numerous near-contacts in a highly cluttered scene. **(b)** From the same episode, the grasped cereal box comes into contact with a foreground can in the cluttered scene. **(c)** A can is stacked on top of another in a cluttered scene **(d)** A significantly occluded grasped sugar box pushes a can across the table

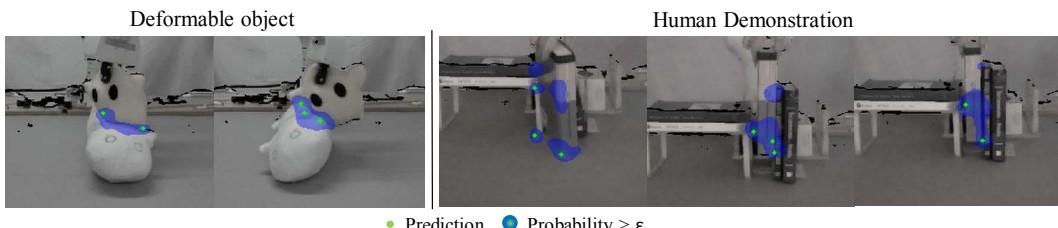

Figure 8: `im2contact` predictions on **(Left)** interactions between two moderately deformable plushies **(Right)** human video of inserting a book into a tight space

grate F/T sensing in Appendix D validate that F/T sensing can improve performance in sim, but will require substantial effort to transfer to real which we hope to actualize in future work.

## 6 Conclusion

We present a method that enables extrinsic contact localization from vision alone with minimal prior knowledge or explicit assumptions on the grasped object and environment. By incorporating well-motivated inputs to our model in simulation, we show successful sim-to-real transfer of our model whereas a naive baseline fares poorly. The method also shows promise on our chosen out-of-distribution settings that include deformable objects and human demonstrations. In future work, we hope to address the limitations of `im2contact` and explore its downstream utility in enabling control tasks involving tool use or assembly.

**Acknowledgments**

We would like to thank Denny Cao for his contributions to the synthetic data generation pipeline and teleoperation interface, as well as the reviewers for their insightful feedback during the rebuttal period. Leon Kim was supported by the NSF GRFP, Yunshuang Li by the Chiang Chen Overseas Fellowship, Michael Posa by NSF CAREER Award FRR-2238480, and Dinesh Jayaraman by NSF CAREER Award 2239301.

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

# A  Randomized Simulation Data Generation

**Environment objects:** At the beginning of each 15 second episode, 8 objects are spawned into the environment with either box or cylinder geometry with equal probability. We do not include spherical geometries as they are rare in household/kitchen settings, though we do introduce them into the possible grasped object geometries.

The spawn position of each object is sampled uniformly in a box above the table ($x$: [0.2m, 0.8m], $y$: [−0.38m, 0.38m], $z$: [0.1m, 0.4m]). Euler angles are sampled uniformly between $\pm\pi$. The respective dimensions for each primitive geometry (height, width, length for box, diameter and length for cylinder, and diameter for sphere) is sampled uniformly between 0.02m and 0.3m, mass sampled uniformly between 0.05kg and 2kg, and friction sampled uniformly between 0 and 1.

**Grasped object:** The grasped object is spawned with the same procedure used to generate the geometry, dimensions, mass, friction as above, but the position is sampled uniformly in a cylinder defined in the end-effector frame with radius 0.001m and end-points at −0.007m and 0.001m along the z-axis. The orientation is sampled uniformly in a cone about the z-axis of the end-effector frame with an aperture angle of $0.7\pi$.

**Robot policy:** Desired delta end-effector positions are sampled at 50Hz from an Ornstein-Uhlenbeck (OU) process which is tracked by an impedance controller. Desired orientation is constant and chosen to match the world frame, though we use very low orientation stiffness in the impedance controller to encourage diverse orientations during contact.

In the general form of the OU process: $dx_t = \theta(\mu - x_t)\,dt + \sigma\,dW_t$, the first term is deterministic and draws the process back to a constant $\mu$ (referred to as "drift") with linear gain $\theta$, while the second term is the stochastic wiener process where the variance is scaled by $\sigma$.

The desired $x, y$ trajectory is sampled independently with different parameters from the desired $z$ trajectory as we'd like to keep the motion in the xy-plane as diverse as possible, while ensuring the end-effector is close enough to the table to make frequent contact with the environment objects and table surface.

For the $x, y$ process, we define four halfspaces in polar coordinates that contribute to the first drift term in the OU process to keep the desired trajectories within a polar rectangle around the robot's workspace. These terms only become active when the end-effector leaves the respective halfspace. Hence, when the robot is within all halfspaces, the $x, y$ process becomes simply a wiener process. The polar rectangle boundaries are defined such that the radius is between 0.35m and 0.7m and the angle is between $\pm 2.2$ radians. For $\theta$ we choose both $x, y$ to be 20 and for the variance matrix we choose a diagonal with both elements equal to $0.2^2$

For the $z$ process, we choose a drift that is 0.2m above the table surface, $\theta$ equal to 1, and a variance of $0.05^2$

# B  Adapted Greedy NMS Algorithm

---

**Algorithm 1** Adapted NMS Algorithm

---

**Input:** $P \in [0, 1]^{H \times W}, p_t \in [0, 1], r_{nms} \in \mathbb{R}^+, K_{iter} \in \mathbb{N}$
**Output:** $\mathcal{C} \subset \mathbb{N}^2$

1  $\mathcal{C} \leftarrow \{(i, j) \mid P_{i,j} > p_t\}$
2  $\mathcal{C} \leftarrow \textbf{sort}(\mathcal{C}, \textbf{by} = P_{i,j}, \textbf{order} = \textbf{descending})$
3  **for** $k = 1, \dots, \min(K_{iter}, |\mathcal{C}|)$ **do**
4      $\mathcal{C} \leftarrow \mathcal{C} \setminus \{(i, j) \mid \left\| ((i - i_k)^2, (j - j_k)^2) \right\|_2 < r_{nms} \textbf{ and } P_{i,j} < P_{i_k,j_k}\}$
5  **end for**
6  **return** $\mathcal{C}$

---

# C Tabular Results

| method | Recall ↑ | Precision ↑ | F1 ↑ | Avg. TP distance ↓ |
|---|---|---|---|---|
| U-Net depth | $0.577 \pm 0.012$ | $0.524 \pm 0.014$ | $0.548 \pm 0.003$ | $3.317 \pm 0.052$ |
| w/o obj-ref+flow | $0.585 \pm 0.011$ | $0.467 \pm 0.008$ | $0.519 \pm 0.004$ | $3.42 \pm 0.08$ |
| w/o obj-ref | $0.559 \pm 0.01$ | $0.5 \pm 0.002$ | $0.528 \pm 0.005$ | $3.341 \pm 0.048$ |
| w/o flow | $0.574 \pm 0.011$ | $0.518 \pm 0.01$ | $0.544 \pm 0.004$ | $3.453 \pm 0.029$ |
| w/o crop | $0.551 \pm 0.011$ | $0.53 \pm 0.015$ | $0.539 \pm 0.005$ | $3.265 \pm 0.061$ |
| im2contact | $0.613 \pm 0.017$ | $0.514 \pm 0.017$ | $0.558 \pm 0.005$ | $3.526 \pm 0.05$ |

Table 1: Metrics on simulation data with all ablations

| method | Recall ↑ | Precision ↑ | F1 ↑ | Avg. TP distance ↓ |
|---|---|---|---|---|
| U-Net Depth | $0.41 \pm 0.022$ | $0.36 \pm 0.017$ | $0.38 \pm 0.01$ | $4.1 \pm 0.138$ |
| w/o obj-ref+flow | $0.58 \pm 0.018$ | $0.44 \pm 0.014$ | $0.5 \pm 0.009$ | $4.09 \pm 0.109$ |
| w/o obj-ref | $0.57 \pm 0.009$ | $0.47 \pm 0.006$ | $0.52 \pm 0.006$ | $4.07 \pm 0.103$ |
| w/o flow | $0.56 \pm 0.019$ | $0.5 \pm 0.011$ | $0.53 \pm 0.005$ | $3.92 \pm 0.085$ |
| w/o crop | $0.44 \pm 0.025$ | $0.39 \pm 0.025$ | $0.41 \pm 0.007$ | $4.47 \pm 0.065$ |
| im2contact | $0.56 \pm 0.022$ | $0.5 \pm 0.017$ | $0.53 \pm 0.005$ | $4.08 \pm 0.094$ |

Table 2: Metrics on real data with all ablations

# D Preliminary Evaluations of Adding Force-Torque Sensing

We have found in our initial efforts to integrate force-torque sensing that this does indeed improve performance in simulation, with an increase in F1 score from 0.56 to 0.60 as seen in Table 3. However, sim-to-real transfer is challenging because of calibration errors, drift, and deterioration. To account for this, we compensate the available joint-torque sensing with a model of the robot together with estimated grasped-object mass to approximate the external joint-torques which can be attributed to contact. However, unmodeled effects remain: robot joint friction is difficult to identify well, and object inertial properties are only coarsely estimated. As a result, im2contact + F/T performs worse in real data: F1 score drops from 0.53 to 0.50 (Table 4). We show qualitative video examples on real data comparing `im2contact` with the addition of F/T sensing at our website: https://sites.google.com/view/im2contact.

We may conclude that F/T sensing does not offer a simple silver bullet solution for consistently improving extrinsic contact sensing in our settings. Incorporating additional sensors to improve the performance of our vision-only `im2contact` approach will require non-trivial additional contributions.

| method | Recall ↑ | Precision ↑ | F1 ↑ | Avg. TP distance ↓ |
|---|---|---|---|---|
| im2contact+F/T | $0.62 \pm 0.004$ | $0.584 \pm 0.011$ | $0.601 \pm 0.006$ | $3.507 \pm 0.042$ |
| im2contact | $0.613 \pm 0.017$ | $0.515 \pm 0.018$ | $0.558 \pm 0.005$ | $3.526 \pm 0.05$ |

Table 3: Metrics on simulation data comparing im2contact with and without F/T sensing

| method | Recall ↑ | Precision ↑ | F1 ↑ | Avg. TP distance ↓ |
|---|---|---|---|---|
| im2contact+F/T | $0.479 \pm 0.024$ | $0.54 \pm 0.017$ | $0.504 \pm 0.011$ | $4.069 \pm 0.123$ |
| im2contact | $0.56 \pm 0.022$ | $0.501 \pm 0.017$ | $0.526 \pm 0.004$ | $4.078 \pm 0.093$ |

Table 4: Metrics on real data comparing im2contact with and without F/T sensing

### D.1 Implementation Details of Adding Force-Torque Sensing

The Franka Panda robot provides estimated external joint-torques by compensating the joint-torque measurements with an internal model of the robot's inertial and kinematic properties. These are then transformed into a "virtual" external force-torque measurement at the end-effector by applying the pseudo-inverse of the jacobian, followed by alignment into the world-frame.

We attempt to compensate the unknown grasped-object's inertial effects by coarsely estimating the induced gravitational wrench. On real data, we assume the robot does not move at the beginning of the episode and average the first 0.5 seconds of the world-frame z-component of the external wrench, followed by division by gravitational acceleration to obtain an estimated mass. In sim, we simply use the ground-truth object mass. In both sim and real, We assume the CoM location is fixed and located 0.15m along -z-axis of the end-effector frame to obtain the adjoint map to compute the approximate gravitational wrench as a function of the end-effector pose.

On real data, we apply a causal low-pass Butterworth filter to reduce the observed force-torque oscillations during free-space motion that we suspect are attributable to coupled effects between the unmodelled joint-friction and the impedance controller.

We integrate the current-most external wrench estimate to im2contact by first passing the 6-dimensional wrench vector through a small MLP, followed by tiling and concatenation to the $15 \times 20 \times 1024$ bottleneck of the U-Net. We train with the same training procedure, hyperparameters, and dataset as before.

## E  Implementation Details of Human Demonstration Evaluations

We localize the human hand in the image by affixing a passive reflective ball to the hand which can easily be thresholded and localized in the RGB-D camera's infrared image stream. We apply the same $90 \times 110$ cropping window to the ball pixel coordinate with a relative shift downward by 55 pixels.

We modify the cropping window hyperparameter during training of im2contact by additionally shifting the window down by 14 pixels to mitigate effects of domain shift in the agent's end-effector. Otherwise, we retain the same training procedure, hyperparameters, and dataset as before.

