# OpenReview forum: "Im2Contact: Vision-Based Contact Localization Without Touch or Force Sensing"
_robot-learning.org/CoRL/2023/Conference — CoRL 2023 Poster_

### Official Review · Reviewer_rgiT · 2023-07-18

**Confidence:** 4
**Originality:** Fair
**Technical Quality:** Fair
**Clarity Of Presentation:** Fair
**Impact:** 3

**Recommendation:**

Weak Accept: I recommend accepting the paper, but will not argue for my recommendation if the majority of other reviewers have a different opinion.

**Review:**

This paper discusses an interesting topic, can a learning algorithm "guess" where and when contact happens, similar to humans watching a video and predicting if contact occurs based on prior knowledge or experience of environmental physics? The general idea is fine based on the close resemblance to human prediction and the results presented. It seems that based on the authors' claim that "yet robots today do not have any approach to reliably sense contacts in general settings" and the method presented in this work, this paper is proposing a solution that contacts estimation from a purely vision-based approach is what this paper is trying to promote for contact-based robot learning as a future direction. I think the statement itself is wrong (we have force sensors commercially available on robots, tactile arrays, or e-skins capable of distributed contact sensing, etc., which is abundantly available in the current literature). The problem is probably related to a disentanglement between the development of contact sensors and how they can be integrated with robotic solutions in applications instead of completely disapproving of this research direction.

The fundamental logic behind the proposed method still relies on the underlying physics of contact for collecting training data, either from simulations built based on such logic or from the reality that the authors gathered from the FT sensors from the robot's wrist. The vision-based observation serves only as an external view easily influenced by the lack of local contact or anomalies from data trained with "common" scenarios. For example, what if there are N number of objects placed inside a bowel, can the proposed method predict contacts using external images such as those in Fig. 1 to make the prediction using a spatula controlled by a robot while moving around? The limitation of this proposed method is obvious by nature, and the claim made by the authors is way too broad and affirmative which is just surprising.

It seems Sim-2-Real transfer is what the authors are really trying to demonstrate as a superior performance of this proposed method, which I think is mostly based on simplified task scenarios and structured settings that the model overfits for overconfident results. As I mentioned, what if placing a few objects inside the bowel or something behind the object with image occlusion and will the method still work? I think this method is fundamentally limited in application scenarios that do not fit a generalizable contact localization close to reality, where things like occlusion happen (almost) all the time.

And based on the results presented, I don't see a clear classification or recognization of contact type identification for point, line, or planar; sticking, sliding, or rolling; single or multiple. In other words, I think the results presented in this work can be resolved by adding or choosing a suitable force sensing technology at the fingertip or object surface and identifying using physics-based analysis of the contact nicely.

The images taken for this paper are hard to see with the limited resolution.

**Quality Of The Limitations Section:**

Limitations are not well addressed

**Questions For Rebuttal:**

1. "robots today do not have any approach to reliably sense contacts in general settings." This statement is a bit absolute and harsh. Can the authors clarify what you mean by general settings? How reliable do you think that is necessary? Is it actually the case that we "do not have any approach" for sensing contacts?

2. The authors claim that the proposed method shows a promising application for "versatile general-purpose contact perception from vision alone" ... "and even under occlusions in its camera view." Can the method estimate contacts when a few balls are placed inside the bowel while conducting experiments similar to those in Fig. 1?

3. Can the authors provide a more realistic application of the proposed method with a more detailed description that it can perform superior that using existing methods or force sensing technology? And can the authors also demonstrate scenarios where the proposed method may fail but the existing force sensing technology still works?

4. Can the authors comment on whether we should move towards a direction where vision-base method only should be used for contact estimation or if the force sensors could still be of any use in the future of contact-related robot learning research and applications.

**Robotics Focus:**

Sufficient demonstration on hardware

**Summary Of Paper:**

This paper presents a vision-based method for estimating contact localization without force sensing based on the motivation that robots today do not have any approach to reliably sense contacts in general settings due to the limitations in sensing technologies or sensor drifts. The proposed method relies only on a single RGB-D camera view of the robot's workspace and learns to identify when and where contact exists. Using simulated data for training, the authors proposed a method that transfers well to an actual robot with data, demos, and results presented in this paper and on a public website.

**Summary Of Recommendation:**

I do not recommend this paper due to the fundamental limitations of the proposed method and the limitations presented in this work. It could be helpful when the scenarios are simple enough or highly structured, which can be alternatively solved using the proposed method for automation purposes or cost consideration. But if any complexity is involved, the method would fail badly due to a lack of local contact sensing with physics-based training data and scenarios, which is fairly limited in applications.

Updated recommendation:
After reviewing the rebuttal made by the authors, I would like to update my recommendation to weak accept. The paper is technical alright, and the idea is interesting in general. I would request the authors to revise statements such as "yet robots today do not have any approach to reliably sense contacts in general settings" and similar others. Also, the authors should strengthen the motivation of this paper so that it reflects a closer connection to robotics.

---

### Official Review · Reviewer_25ZM · 2023-07-20

**Confidence:** 4
**Originality:** Good
**Technical Quality:** Good
**Clarity Of Presentation:** Very Good
**Impact:** 3

**Recommendation:**

Weak Accept: I recommend accepting the paper, but will not argue for my recommendation if the majority of other reviewers have a different opinion.

**Review:**

I have no problem with the quality or clarity of the writing or presentation. I just have issues with the motivation and the use cases of this proposed method of detecting contacts from a single view that is directly in front of the robot. In addition, there are many false positives still because only one camera is used. I do think the use of simulation is good for collecting ground truth contact data, but the data seems to have only been collected from one view. My main review is contained in the questions for rebuttal and the summary of recommendation.

**Quality Of The Limitations Section:**

Limitations are addressed clearly

**Questions For Rebuttal:**

1. I agree with your limitation that in not using the robot's proprioceptive feedback such as forces and torques, you could reduce the number of false positives I saw in your supplementary videos. It would be great if you could give some initial results of fusing the robot's force and torque data to improve the real world predictions.
2. I also agree with your limitation that a lot of false positives and contact localization could be improved with more cameras. My question is why you had to put the realsense camera in front of the robot. From your motivation in the introduction, you mentioned wanting to put a book in a bookcase which requires a lot of contact sensing. However, it is unrealistic that in a real domestic mobile robotic setting that you would have a camera always in front of the robot. So how would your method work from around a head mounted view like from the Nimbro avatar robot? (https://www.youtube.com/watch?v=8AwgGSpcAe8)
3. What I find most promising for this work is if it could be applied to human egocentric videos like epic kitchens and ego4d where humans do not have force torque sensors unlike the Franka robot. What do you think are the limitations to making your pipeline work on those videos?
4. Also you should put a table in the main paper and not just on the supplementary website. I think it's easier to read a table than Figure 4.

**Robotics Focus:**

Sufficient demonstration on hardware

**Summary Of Paper:**

The authors aim to sense contacts between a robot and a tool with the environment using a single rgbd camera. They use UNet Depth and cropped images of the robot's tool and optical flow as some of the inputs into their network. To get the training data for their network, they use Gazebo to get ground truth contacts with a variety of primitive objects. They show that some of the baselines and their method are able to have sim 2 real transfer with some hand labeled contacts. In their rebuttal, they added new experiments using a human hand to interact with objects and predicting the contacts as well as using force torque data to reduce the amount of false positives.

**Summary Of Recommendation:**

I have increased my decision from a weak reject to a weak accept due to the new experiments and real world results. While I still believe that this approach of using only a singular rgbd camera to detect contacts between a tool and the environment is not very practical, I think that the idea is still not that bad and could be at least published. As long as the authors can incorporate the results from the rebuttal nicely into the paper, I could argue that it is better than a weak reject but still worse than a strong accept. I still think that in reality multiple cameras could be used to get a nice mesh reconstruction of the environment which would aid in better contact detection "without touch or force sensing". In addition, currently the sim 2 real transfer only works if the camera and robot are only in the same configuration. If the authors could leverage NERF to be view agnostic, it would be great.

---

### Official Review · Reviewer_qKK8 · 2023-07-20

**Confidence:** 4
**Originality:** Good
**Technical Quality:** Good
**Clarity Of Presentation:** Very Good
**Impact:** 3

**Recommendation:**

Weak Accept: I recommend accepting the paper, but will not argue for my recommendation if the majority of other reviewers have a different opinion.

**Review:**

Overall, I found this paper to be somewhat mixed in its efficacy. In general the methods the authors use seem well-posed, and supervising contact location prediction using simulation seems like a reasonable approach. Further while the lack of baselines (since the authors have posed a relatively novel problem) makes it difficult to judge the overall performance of the final model, he contact probability heat maps it returns for novel objects, e.g., Figure 8, seem reasonable, if somewhat messy.

However, since the authors are studying a relatively new problem, I think it is important for them to make the case that this problem is of interest to the broader robotics community. In my opinion, the paper would be greatly improved by a more thorough discussion of the authors' overall motivation -- how does pixel-wise contact prediction connect to canonical manipulation tasks like tool use, object pile manipulation, assembly, etc?

This justification is especially important since other works studying contact prediction, e.g., ContactNets [1] (uncited here), use signed distance representations, rather than predicting contacts in pixel space, which is a more intuitive/useful representation for performing things like in-hand manipulation, etc. I think the paper would also improve from a discussion of the benefits of a pixel-wise contact representation (which has many ambiguities, e.g., if a line contact lies along the camera's view direction) compared to other representations.

These motivational concerns aside, however, I found the paper to be clearly written and the approach sound.

[1] ContactNets: Learning Discontinuous Contact Dynamics with Smooth, Implicit Representations, Pfrommer et al., CoRL '20.

**Quality Of The Limitations Section:**

Additional details required

**Questions For Rebuttal:**

Can you discuss more the practical usefulness of a pixel-wise contact representation? What manipulation problems do you think this contact/shape representation would be useful for? Is it easy to generalize to multiple objects?

Do the ablation studies in Fig. 4 show that cropping is really the only added feature that improves performance significantly? Why include the additional information/complexity required (e.g., optical flow) if the performance gains are relatively small?

**Robotics Focus:**

Sufficient demonstration on hardware

**Summary Of Paper:**

This paper studies the problem of vision-only contact prediction. The authors propose to use a variant of a U-Net to predict pixelwise probabilities of a contact between an object held by a robot and the environment. To supervise this network, they use contact point labels generated in a simulator (Gazebo) and modify the architecture slightly (providing context views, cropping the image to only include the object, and providing optical flow data) to improve performance. While training on simulated contact data raises a significant sim-to-real issue, the authors study their method's performance on real objects and observe relatively good performance.

**Summary Of Recommendation:**

While I think the results of the paper are sound and correct, I have pretty serious concerns about the motivation -- i.e., it's unclear why predicting pixel-wise contact probabilities is useful for downstream robotics tasks. Unfortunately, many of the contributions of this work are closely connected to the pixel-wise representation and don't necessarily generalize to other representations, such as Signed Distance Functions (SDFs).

However, with either a firm rebuttal from the authors or good discussion from other reviewers, I am open to having my mind changed.

Post-rebuttal edit: My mind was changed. I think the authors can/will address my concerns, and found the additional human experiments to be a very good example of how the representation is useful/generalizable. While I personally am skeptical about the longer-term usefulness of the result, I think it's an important direction to explore and the results are surprisingly good for the simple setup.

---

### Official Review · Reviewer_PPMJ · 2023-07-23

**Confidence:** 4
**Originality:** Very Good
**Technical Quality:** Very Good
**Clarity Of Presentation:** Very Good
**Impact:** 3

**Recommendation:**

Weak Accept: I recommend accepting the paper, but will not argue for my recommendation if the majority of other reviewers have a different opinion.

**Review:**

## Strengths

This paper develops an interesting question to the challenging problem of estimating extrinsic contacts solely from images. The paper is very well-written, easy to follow, and describes the method in sufficient detail to allow a skilled practitioner to re-implement it. As such, it is extremely relevant to the robot learning, and particularly, the robot manipulation communities.

A lot of care is taken to ensure the method generalizes to real-world scenarios with non-trivial extrinsic contact. While a simulation environment allows for generating as much synthetic data as possible, the paper demonstrates that training a single model on simulated RGB-D images alone does very poorly when applied in the real world. Much of the design decisions surrounding im2contact are substantiated by the experimental results in Fig. 4 and the tables on the supplemental webpage.

The qualitative results include a number of interesting scenarios and a set of complex objects. This also includes an out-of-distribution set (although I'd presume this is cherry-picked), of the approach showing some transferability to deformable objects, or rigid objects with highly non-convex shapes.

Overall, the central idea is sound; the experiments are well-executed; and the paper is well-written.


## Weaknesses

There are, however, a few edits I would like to see in the paper, which should further improve its readability; and make it self-contained.

**Discrepancies between Fig. 4 and suppl. tables**: The precision-recall plots in Fig. 4 of the paper seem to have slightly different numbers compared to the tables at the bottom of the supplemental webpage. While the trends are still consistent across them, it will be important to know what causes this discrepancy (perhaps that the experiments each use a different random seed?) Further, moving the real-data table to the main manuscript will make the paper self-contained.

**Impact of optical flow**: The significance of using the optical flow has not been sufficiently established in the main manuscript. The supplemental webpage presents additional tables where the variant w/o flow is worse than the full method (im2contact); but since this is claimed as being essential to improve performance, I would rather have preferred to see this table in the main paper. I would also have liked to see a more nuanced discussion into how optical flow helps (i.e., what kinds of scenarios), and if it hurts in some cases. (currently Fig. 6 is the only anecdotal evidence in support of using flow information)

**Discussion of limitations / assumptions**: im2contact presents an interesting solution to extrinsic contact localization; and the paper thoughtfully discusses some of the practical limitations (contact localization works only in 2D; is noisy; vision-only). A more fundamental assumption underlying im2contact is that extrinsic contacts aren't heavily occluded. However, in practical contact-rich manipulation scenarios (e.g., peg / cable insertion), the grasped object is often self-occluded by the manipulator, which makes 'obj-ref' difficult to specify; limiting the applicability of im2contact.

**Alternative obj-ref schemes**: The design space for specifying the reference object seems a bit under-explored. Currently, 'obj-ref' is an additional reference depth image of the grasped object. Are there any exclusive benefits to using this representation, opposed to alternative choices (e.g., heatmap, region of interest, bounding box, etc.)?

**Quality Of The Limitations Section:**

Limitations are addressed clearly

**Questions For Rebuttal:**

I would like to see the questions I raised in the "weaknesses" section of my review discussed.

**Robotics Focus:**

Sufficient demonstration on hardware

**Summary Of Paper:**

im2contact is an approach to estimate extrinsic contacts (i.e., contact between an object held by a manipulator with the environment) by looking at an RGB-D third-person view of the robot and its workspace, and additional cues such as optical flow and reference images of the grasped object. The approach is trained in simulation and deployed on a real-world robot manipulator (a tabletop workspace).

**Summary Of Recommendation:**

The paper presents an interesting method for extrinsic contact localization from RGB-D images. The proposed approach is sound and interesting; the paper looks well-executed. There are a few (minor) clarifications I would like to see discussed over the author response and revision phase. This is an interesting initial system with promising demonstrations for contact localization; however to be generally applicable across a wide range of manipulation tasks (esp ones that are contact rich), several issues need further addressal. For these reasons, I will score this a "weak accept".

---

### Author Response · Authors · 2023-08-11
**Global Response**

We thank the reviewers for their thorough feedback and, in advance, for their continued efforts during this rebuttal period. Here, we discuss high-level points of emphasis common to multiple reviews. We provide additional responses to individual reviewers in separate responses.

**1. Motivation and quality of results**

To state our motivation with renewed emphasis, we aim to localize contact between a grasped object and the environment, referred to as extrinsic contact. We seek to do this online from RGB-D inputs, without prior knowledge of the grasped object or any objects in the scene. Such information is valuable in manipulation settings involving rich extrinsic contact interactions, particularly tool-use and assembly [1,2].

Our work demonstrates a previously unexplored paradigm for contact sensing: we show, for the first time, that it is possible to directly extract contact information with a single, inexpensive depth camera. In doing so, we remove many of the limiting assumptions of prior work (see Sec 2), enable exclusively training in simulation where contact labels are easy to obtain, and produce reasonable contact estimates between unknown grasped and environment objects. No prior work has tackled these challenging settings.

On a diverse real-world dataset spanning point, line, and surface contacts (e.g. Fig 7 and associated [videos](https://sites.google.com/view/im2contact/videos/full-ablation-grid)), varied real-world object types (e.g. Fig 8), and occlusions of the grasped and other objects (e.g. Fig 5), im2contact promotes the baseline up from an F1 score of 0.36 up to 0.53 and attains an average of 1.2 cm of error from our annotations at a nominal depth of 1.2m (see [Supp. Table](https://sites.google.com/view/im2contact/home)). Furthermore, the predicted 2D heatmaps reasonably coincide with our manual labels, even in the presence of moderate occlusions of the grasped object with a foreground object, where human annotators required multiple camera views.

[1] Van der Merwe, Berenson, and Fazeli, “Learning the Dynamics of Compliant Tool-Environment Interaction for Visuo-Tactile Contact Servoing.” CoRL 2022.

[2] Morgan et al., “Towards Generalized Robot Assembly through Compliance-Enabled Contact Formations.” ICRA 2023.

**2. Utility of pixel-wise contact representation**

Prompted by reviewer qKK8’s request for a discussion of the utility of our pixel-wise extrinsic contact representation, we point to similar works for compelling demonstrations of the applications of pixel-wise contact maps for accelerating robot policy learning [3] and in a visual-servoing framework [4]. In addition, we believe im2contact’s outputs are readily integrated with other image-based features which are well-established in the robot learning literature. Affordance maps, keypoints, correspondence maps, and segmentation masks [5,6,7,8] have all demonstrated utility in various robot manipulation contexts and we believe combining these with im2contact opens up new and exciting research directions. Furthermore, new experiments in our fourth point below validate the feasibility of using im2contact to transfer contact information from human demonstrations to robots.

[3] Bahl et al., “Affordances from Human Videos as a Versatile Representation for Robotics.” CVPR 2023.

[4] Collins et al., “Visual Contact Pressure Estimation for Grippers in the Wild.” IROS 2023.

[5] Zeng et al., “Robotic Pick-and-Place of Novel Objects in Clutter with Multi-Affordance Grasping and Cross-Domain Image Matching.” IJRR 2019.

[6] Qin et al., “KETO: Learning Keypoint Representations for Tool Manipulation” ICRA 2020.

[7] Florence, Manuelli, and Tedrake, “Dense Object Nets: Learning Dense Visual Object Descriptors By and For Robotic Manipulation.” CoRL 2018.

[8] Gualtieri and Platt, “Robotic Pick-and-Place With Uncertain Object Instance Segmentation and Shape Completion.” RAL 2021.

---

> ### Author Response · Authors · 2023-08-11
> **Global Response (cont.)**
>
> **3. Why restrict im2contact to a single RGBD camera, and experiments with F/T**
>
> We agree with reviewers qKK8, 25ZM, and rgiT that incorporating additional camera views or force sensing inputs could improve our method’s performance; this is discussed in Sec 5. Indeed, most prior efforts towards extrinsic contact estimation have used F/T sensors on the robot to indirectly sense extrinsic contacts, but this remains a difficult problem. As such, prior works have relied on strong assumptions such as known object geometries, fixed object categories, or limited contact configurations (see Sec 2). In this paper, our focus is on demonstrating the feasibility of vision-only extrinsic contact localization with minimal assumptions.
>
> When more sensors are available, the additional information they provide should indeed improve im2contact’s performance in principle. We have found in our initial efforts to add F/T sensors (as requested by reviewer 25ZM) that this does indeed improve performance in simulation (F1 score: 0.56 -> 0.60). However, sim-to-real transfer is challenging because of calibration errors, drift, and deterioration in real F/T sensors (See L 36-38). To account for this, we compensate the available joint-torque sensing with a model of the robot together with estimated grasped-object mass to approximate the external joint-torques which can be attributed to contact. However, unmodeled effects remain: robot joint friction is difficult to identify well, and object inertial properties are only coarsely estimated. As a result, im2contact + F/T performs worse in real data: F1 score drops from 0.53 to 0.50. We show qualitative examples in the attached zip file, and for convenience, also display them on the anonymous [URL](https://sites.google.com/view/im2contact/rebuttal) included in the paper.
>
> We may conclude, in accordance with prior works, that F/T sensing does not offer a simple silver bullet solution for consistently improving extrinsic contact sensing in our settings. Incorporating additional sensors to improve the performance of our vision-only im2contact approach will require non-trivial additional contributions.
>
> **4. New results: demonstration on human video**
>
> As we have stated above and in the paper, our vision-only approach is intended to minimize assumptions and thereby apply to domains well beyond where extrinsic contact sensing has thus far been possible.
>
> Reviewer 25ZM suggests a particularly intriguing setting that we have now tested: extracting contact from human demonstrations. We find that im2contact continues to work well on videos of humans performing various tasks, including bookshelf insertion (motivated in Sec 1), manipulating objects in a bowl with a spoon (suggested by reviewer rgiT), shuffling items on a dish rack, and writing/erasing on a whiteboard. These examples are included in our attachment and [website](https://sites.google.com/view/im2contact/rebuttal).
>
> We are excited about these human demonstration results since this enables another use case in addition to those we have mentioned above: guiding robot policy learning from human data! For example, Bahl et al. [3] utilizes learned 2D intrinsic contact maps of human hand interactions to accelerate imitation learning. Note that we made virtually no changes to our approach for these results: we still train im2contact in simulation with a Franka Panda robot as before, but crop the image inputs marginally lower than before, to minimize how much of the robot / human hand is visible in sim / real videos to minimize domain shift effects.

---

### Decision · Program_Chairs · 2023-08-30

**Decision:**

Accept (Poster)

**Comment:**

## Summary of Paper
The paper proposes a method for detecting contacts purely based on RGB-D input. The approach is shown to work on real robot data even with occlusions, with various contact types and objects.

## Summary of Reviews
The paper is well written with a compelling idea and method. The experiments are very well executed. Some details needed to be clarified. There were some doubts about the performance. The major concern of several reviewers circled about the question if this method is useful in practice.

## Influence of Rebuttal
The authors nicely argued for why the method is practically relevant. The additional experiments (human video & F/T inclusion) were well received. The quality of the results was well explained.
We had some discussions between the reviewers and AC. In the end also Reviewer rgiT was convinced that the paper is a good fir for CoRL (see updated review). We now see potential application fields for vision-only contact detection in robotics, possibilities to use this method in sensor fusion, we don't think the many false positives are a deal-breaker.

## Suggestions for Improvement
Please include all the promised changes and the new results in the final version. In particular please revise the (by your own admission poorly worded) statements about contact sensing in robotics.